# Pluripotential GluN1 (NMDA NR1): Functional Significance in Cellular Nuclei in Pain/Nociception

**DOI:** 10.3390/ijms241713196

**Published:** 2023-08-25

**Authors:** Terry A. McNearney, Karin N. Westlund

**Affiliations:** 1Department of Neuroscience and Cell Biology, University of Texas Medical Branch Galveston, Galveston, TX 77555-1043, USA; tmcnearn@yahoo.com; 2Department of Internal Medicine, University of Texas Medical Branch Galveston, Galveston, TX 77555-1043, USA; 3Department of Microbiology and Immunology, University of Texas Medical Branch Galveston, Galveston, TX 77555-1043, USA; 4Department of Anesthesiology, University of New Mexico Health Sciences Center, Albuquerque, NM 87131-0001, USA; 5Biomedical Laboratory Research & Development (121F), New Mexico VA Health Care System, Albuquerque, NM 87108-5153, USA

**Keywords:** pain, nuclear translocation, nucleus, nucleolus, epigenetics, membrane trafficking, inflammation, glutamate, central sensitization, mood, nucleolar stress, memory

## Abstract

The *N*-methyl-D-aspartate (NMDA) glutamate receptors function as plasma membrane ionic channels and take part in very tightly controlled cellular processes activating neurogenic and inflammatory pathways. In particular, the NR1 subunit (new terminology: GluN1) is required for many neuronal and non-neuronal cell functions, including plasticity, survival, and differentiation. Physiologic levels of glutamate agonists and NMDA receptor activation are required for normal neuronal functions such as neuronal development, learning, and memory. When glutamate receptor agonists are present in excess, binding to NMDA receptors produces neuronal/CNS/PNS long-term potentiation, conditions of acute pain, ongoing severe intractable pain, and potential excitotoxicity and pathology. The GluNR1 subunit (116 kD) is necessary as the anchor component directing ion channel heterodimer formation, cellular trafficking, and the nuclear localization that directs functionally specific heterodimer formation, cellular trafficking, and nuclear functions. Emerging studies report the relevance of GluN1 subunit composition and specifically that nuclear GluN1 has major physiologic potential in tissue and/or subnuclear functioning assignments. The shift of the GluN1 subunit from a surface cell membrane to nuclear localization assigns the GluN1 promoter immediate early gene behavior with access to nuclear and potentially nucleolar functions. The present narrative review addresses the nuclear translocation of GluN1, focusing particularly on examples of the role of GluN1 in nociceptive processes.

## 1. Introduction

The *N*-methyl-D-aspartate (NMDA) glutamate receptors function as plasma membrane ionic channels and are part of very tightly controlled cellular processes activating neurogenic and inflammatory pathways. A large body of studies supports their functions in both normal physiology and disease states with a focus on the NMDA NR1 subunit (new terminology GluN1 subunit [1]) function in psychiatric and neurologic conditions. NMDA receptors are activated by excitatory amino acid (EAA) agonists, glutamate, aspartate, and NMDA. NMDA receptors in low levels are essential for neuronal development, differentiation, learning, survival, and plasticity [2,3]. NMDA receptors play a critical role in pre- and post-synaptic plasticity, especially learning and memory [4]. However, when glutamate receptor agonists are present in excess, binding to NMDA receptors produces neuronal/CNS/PNS excitotoxicity, pathology, conditions of acute pain [5,6], and ongoing severe, intractable pain [7,8]. Conditional deletion of the GluN1 subunit in the spinal cord dorsal horn reduces injury-induced pain [9]. Beyond this is involvement in anxiety/depression [10], disease states (schizophrenia, Parkinson’s [11,12]), dementia [13,14,15], and seizures [16,17]. 

Earlier studies established that the GluN1 subunit anchor component was necessary for heterodimer formation, cellular trafficking, and nuclear localization that was functionally specific [18,19]. Four subunits assemble the glutamate receptors, and each heterodimer strictly requires the GluN1 subunit to anchor GluN2 or GluN3 subunits, comprising a functional ionic channel. GluN1 subunits bind combinations of GluN2A, 2B, and GluN3A and 3B subunits that form heterodimers to anchor these components on the cellular plasma membrane [20,21,22]. The binding composition is noted as two obligatory GluN1 subunits, of which there are now reported at least eight distinct variant subunits and four or two variable subunits from the GluN2 (GluN2A-2D) and GluN3 (GluN3A-3B), respectively, producing 8 × 6 = 48 potential heterodimers.

The GluN1-1a subunits have a nuclear localization sequence (NLS) exclusive of tissue specificity [18,23]. Additional studies have reported the alternative splicing of messenger RNA producing similar proteins that target different tissues, determine its functional fate [24], and have expanded studies to include expression and functioning in non-neuronal tissues. Much of the recent research has focused on characterizing the GluN2 subunits of the GluN1/GluN2 heterodimer. Shifts in the composition of the heterodimer, based on GluN2 or N3 subunits, denote the functional roles of the receptor complex and promote the neuroplasticity of the glutaminergic system throughout the CNS [23,24].

Over 2800 articles are cited in PubMed pertaining to GluN1 subunits from 1990 to 2023. It should be noted that many of the citations in this work are reviews of the literature. Close to 800 report information on GluN1 subunit expression and function in human tissue. Many of the described cellular processes are in the context of inflammatory, ischemic, and/or neurogenic (nociceptive) milieux relevant to pain states. Likewise, increased GluN1 subunit staining occurs in activated non-neuronal cells, such as epithelial cells [25]; lymphocytes [26], macrophages, type A and B synoviocytes [27,28]; megakaryocytes [29,30], osteocytes, osteoblasts and osteoclasts [31].

Here, we review the GluN1 subunit containing subnuclear organelle organization and ribosomal DNA arrays regulated internally or by epigenetics and how their interactions impact cell nuclear structures and downstream implications for cellular function first introduced in 2015 [32,33]. We will build on current hypotheses for some of these regulatory and pathologic functions based on reported subnuclear interactions and immunochemical imaging. Much has been parsed and gleaned from published reports of GluN1 subunit binding to functional transcriptional sites and subsequent neuronal responses. The reader is also directed to several published reviews [3,5,12,34,35,36,37].

## 2. Potential Fates of GluN1 Subunit in the Cellular Nucleus 

### 2.1. GluN1 Subunit Signaling Induces Nuclear Translocation 

GluN1 subunit occupation in the cell nucleus has been reported since early 2000, and a resurgence of research is providing a better understanding to fill in the gaps. This has included reports of the bipartite NLS contained in the GluN1-1a subunit [18,38], protein isoforms, and sequence cassettes. For example, the GluN1/N2A/B functional tetramer is inserted into the cellular plasma membrane site as an ionic channel. When activated, the heterodimer is phosphorylated at the GluN2 cytoplasmic regions, and the GluN1 subunit transmembrane region protein undergoes intermembrane proteolysis, cleaving the cytoplasmic portion in the cell membrane. The heterodimer internalizes and translocates from the cell’s plasma membrane to the cell’s nuclear membrane [18,35]. This region contains a bipartite NLS with two clusters of short sequences of basic amino acids, mainly lysines (K) and arginines (R), separated by a link of a variable number of amino acids. The two regions of basic amino acids on the protein surface comprise the regions recognized for binding along the nuclear membrane and passing into the nucleus. In the case of GluN1-1a, the NLS sequence regions with basic amino acids in bold type are: 

**KRHK**-spacer region—**KKK**ATFRAITSTLASSF**KRRR** [38].

### 2.2. GluN1 Subunit Staining in the Cellular Nucleus

The nuclear translocation of the GluN1 subunit after glutamate activation suggests it also plays a direct role in the fast intracellular signaling responses to extracellular glutamate activation. Long-term potentiation (LTP) mediated facilitation requires an active importin nuclear import pathway [39]. Examination of the function of the GluN1 subunit in nuclear translocation has found the C1 domain binds calmodulin to assist calcium entry into the nucleus [19,23]. This regulates LTP and synaptic plasticity in hippocampal cultures. Glutamate-mediated (LTP-like) overactivation in pain states likely involves calcium entry into the nucleus through ion channels that include GluN1 subunits. The evidence for this appears below. 

Detection of increased cell nuclear staining of GluN1 subunit in activated cells is easily appreciated in vivo or in vitro. For example, increased cell nuclear staining of the GluN1 subunit was observed with visual microscopic examination of human clonal neuroblastoma cell cultures (SH-SY5Y) activated with glutamate or NMDA for 4 h, and nuclear staining was inhibited by preincubation with active NMDA protein tyrosine kinase (PTK) inhibitors genistein or staurosporin in vitro [40]. Increased nuclear staining was also easily observed by 2 h after incubation with NMDA and ACPD in human synovial fibroblast cells [27]. Additionally, staining of the GluN1 subunit on the nuclear rim and nucleoplasm was appreciated in the presence of cycloheximide [40], demonstrating that nuclear localization is an incitement event for the subunit and not solely the purview of newly synthesized GluN1 subunit. 

In complementary studies inducing synovial joint capsule inflammation, secondary tactile allodynia in the same hindlimb resulted in the same time span of two hours following excess glutamate released into the joint capsule from activated peripheral nerves in a rat intra-articular K/C arthritis model [41]. The afferent nerve and spinal glutamate neurons release glutamate into the spinal cord dorsal horn (SCDH) in response to activation or injury [42]. Models providing information on nociception/pain study either acute pain, where glutamate receptor stimulation is limited timewise, or chronic pain, where the glutamate receptor is overstimulated long-term. Many molecular, epigenetic, and inflammatory events are ongoing, accompanied by prolonged exposure to glutamate and overactivation of glutamate receptors. Continued activation of the glutamate receptors results in excessive intracellular chloride that increases the neuronal membrane potential above the threshold, resulting in the reversal of GABA from inhibitory to excitatory continuous firing, contributing to chronic pain [43,44]. 

Electron microscopic (EM) data demonstrating a shift in the distribution of the GluN1 subunit, which is typically on the cell membrane shifts to a subcellular location in the cytoplasm and nuclear membrane supports NR1’s role as a rapid intracellular mediator acting through direct communication with the nucleus (Figure 1) [38], nucleolus (Figure 2) [45], and is more easily seen in the cytoplasm in cell culture (Figure 3). In addition to increased levels of GluN1 subunit detected on the cellular nuclear membrane, previous studies have demonstrated its subcellular proximity to nuclear pores of the nuclear membrane in rat lumbar spinal cord spinothalamic tract neurons by immunolabeled EM (Figure 1D; Figure 2, long yellow arrows). In the next few hours, increased GluN1 subunit staining was also appreciated throughout the nucleoplasm containing densely and loosely packed chromatin, including the areas proximate to the cell nucleolus (Figure 2, light blue arrows [45]). GluN1-tagged immuno-EM showed GluN1 subunit along the chromatin borders of the cell nucleolus and in the nucleoplasm proximate to the nucleolus (Figure 2, light blue arrows [45]). 

Human clonal SW892 synoviocyte cells activated with NMDA + ACPD also demonstrated GluN1 subunit immunostaining with light microscopic examination (Figure 3 and see [27]). Immunostaining for the GluN1 subunit is evident in naïve cultures only in the cytoplasm (Figure 3A). After activation with NMDA + ACPD, a marked increase in immunolocalization of the GluN1 subunit appears along the nuclear membrane and throughout the nucleoplasm (Figure 3B,C). After 3 h, the GluN1 subunit was localized throughout the nucleoplasm of densely and loosely packed chromatin. Increased staining was evident in the perinuclear region of the cytoplasm, along the borders of the nuclear envelope, and collecting along the edges of the nucleolus. In fact, the activation has induced multiple nucleoli rimed with the GluN1 subunit, indicative of its activated state (Figure 3 inset). When the synovial cultures were pretreated with inhibitor U0126 (5 µM) prior to the treatment with NMDA and ACPD for 3 h, the nuclear and nucleolar translocation was abrogated, including nuclear staining that would indicate nuclear translocation or subnuclear organelle activation. U0126 is a MAPK/ERK kinase and highly selective inhibitor of MEK 1 and 2 activity, that strongly abrogates GluN1 subunit cellular increases. Pretreatment with U0126 was shown previously to protect neurons against oxidative stress [46,47].

## 3. Selected Animal Models of Nociception, Cell Cultures, and Tissues Highlighting In Vitro Cellular GluN1 Subunit Activity 

Changes in GluN1 subunit cellular localization in the spinal cord were followed in a rat arthritis model induced with intraarticular (i.a.) injection of irritants kaolin and carrageenan (K/C) [40]. Immediate dynamic neuronal communication between peripheral joints, the spinal cord, and the consequent central processing in the brain was assessed as changes in the animals’ behavior. By four hours after knee joint inflammatory insult, rats developed increased secondary footpad hypersensitivity to heat tested using the Hargreaves test and paralleled by dorsal horn glutamate release. Previous studies showed dorsal horn glutamate release could be blocked by spinal administration of NMDA receptor antagonist AP7 (60 µM, microdialysis) [48]. Sensitization of dorsal horn neurons was shown previously to be reversed by blocking NMDA receptors [49,50]. Elevated glutamate levels have been measured in the joints of patients with arthritis [51].

Increased GluN1 subunit expression in the lumbar spinal cord dorsal horn was evident ipsilateral to knee joint inflammation in individual neuronal cells with a shift in the concentration of staining from cellular membrane rim to perinuclear and nuclear regions [40,45]. This was determined by immunocytochemical localization with both light and electron microscopy. GluN1 subunit protein expression was increased, determined by Western Blot and immunolocalization utilizing both C- and N-terminal antibodies [40]. 

The effects of PTK inhibition on GluN1 subunit protein expression and subcellular localization were also examined in the acute experimental arthritis model [40]. The PTK inhibitors genistein and lavendustin A were administered into the lumbar spinal cord by microdialysis. The PTK inhibitors reduced pain-related behavior and the occurrence of cellular, histological GluN1 subunit expression that increases in the spinal cord within 4 h after inflammatory insult to the knee joint. Genistein and lavendustin A (but not inactive lavendustin B or daidzein) effectively reduced the shift of the GluN1 subunit from the cell membrane to a nuclear localization. Cycloheximide blocked most of the glutamate-activated upregulation GluN1 subunit content, confirming the synthesis of new protein in response to inflammatory insult contributions, but it was not essential since some nuclear rim staining remained. 

These studies provide evidence that inflammatory activation of peripheral nerves initiates an increase in the GluN1 subunit in the spinal cord coincident with the development of pain-related behaviors through glutamate non-receptor, protein tyrosine kinase-dependent cascades. 

Other studies have reported similar inhibition of GluN1 subunit nuclear translocation using PTK inhibitors in animal or cellular models of depression, anxiety, and Parkinson’s [52,53]. Table 1 lists a brief synopsis of experimental nociceptive/pain models of arthritis, formalin injection, colitis, and inflammatory bowel syndrome (IBS) that directly assess GluN1 subunit expression and function and expand the pain models and conditions. Studies demonstrating increased NMDA receptor agonists or inflammatory mediator levels, and preincubation with enhancing agents are also included. A comprehensive review, including NMDA receptor activation in nociceptive models for orofacial pain, has been recently published [54]. 

## 4. Roles and Potential Fates of GluN1 Subunit in The Cellular Nucleus

The regional and functional assignments provided in the GluN1/2 subunit heterodimers have been attributed mostly to the GluN2 subunits, after they are trafficked and delivered to the nucleus via the GluN1 receptor subunit. However, the GluN1 subunit has several indicators of its own potential to influence heterodimer and/or subnuclear organelle assignments. 

1. The putative NLS and/or nucleolar localization sequences (NoLS) located in cassette 1 of the GluN1 subunit has two short regions rich in basic amino acids, which comprise the NLS and are potentially involved in nucleolar signaling [66,67]. GluN1 subunits are tightly subjected to multiple levels of regulation, affecting subunit expression, subcellular location, and assembly of functional receptors, and their signaling complexes [34].

2. The gene for the GluN1 subunit is expressed in early development in virtually all neurons, and is transcriptionally upregulated during neuronal differentiation. NMDA agonists and EAAs increase cellular GluN1 subunit levels, as demonstrated with widespread cell membrane, intracellular, and nuclear GluN1 subunit staining. GluN1 subunit nuclear translocalization is reported for both human synoviocytes (Figure 3) and for rat spinal cord nociceptive neurons at light and EM levels [40,45]. Nuclear translocation of GluN1 is reported for neurons in eye tissue [68,69]. Additionally, GluN1 subunit activation has been reported in models of ischemia, neurogenic, and inflammatory responses, which also stimulate EAA release and increase GluN1 subunit expression. Zhou and Duan have reported that both GluN1 and GluN2 are needed to translocate [23].

3. There is close coordination in neurons between the assembly of functional heteromeric to tetrameric receptors and the fates of these individual subunits. In addition, two pools of mRNA for the GluN1 subunit have been reported with distinct translational activities. These generate two stores of GluN1 subunits that are differentially assembled with GluN2 subunits to form heterodimers with distinct functions and turnover rates, providing an additional and possibly tissue-specific level of control for protein turnover and trafficking (Figure 4) [20]. Nuclear membrane translocation reportedly occurs by endocytotic and de novo mechanisms. Activity dependent clathrin mediated internalization of GluN1 subunit is reported [35]. Nociceptive neurons are overactivated in pain states with increased GluN1 subunit cellular and nuclear ring immunostaining [40]. Staining is greatly reduced by tyrosine kinase inhibition. Post-transcriptional mechanisms also contribute to GluN1 subunit regulation in brain development [70,71]. Studies have also reported the importance of post-translational histone modifications in epigenetic transcriptional control of nociceptive pathways [32,72]. 

4. Structurally, the GluN1 subunit promoter region is located directly upstream of the transcriptional start site (TSS) and is exceptional for the number of transcriptionally reactive binding regions close to the 3’ site. Transcriptionally active binding regions have been reported for SP-1, NFkB, MEF-2, GC-rich regions, CREB, REI, AP-1, egr-1, and ARC binding regions [5]. The GluN1 promoter allows a spectrum of potential responses, such as activation, de-repression, or suppression of downstream transcribable sequences. 

5. GluN1 subunit activation is reported to consequently activate Immediate Early Genes (IEG), e.g., c-fos, zif268, and egr-1 [73,74], and conversely, IEG products are reported to activate GluN subunits. The term IEG describes viral regulatory proteins or cellular proteins generated immediately following stimulation of a resting cell by internal or external stimuli, triggering immediate gene transcription that does not require de novo protein synthesis. IEG products are usually transcription factors, which are DNA-binding protein activators of signaling pathways. They are rapidly and transiently activated to respond to a plethora of cellular and extracellular stimuli, serving as an important cellular first response system. The GluN1 subunit can direct downstream nuclear functioning via nuclear DNA binding sites, immediate early gene products, cytoplasmic input, and environmental signals. 

## 5. Nucleolar Stress and Pain

Table 2 is a summary of reported locations on subnuclear organelles and interactions of DNA binding regions of the GluN1 subunit in the cell nucleus. Recent studies have described the nucleolus as an important and sensitive stress sensor, acting as a networking or signaling hub that can influence cell fates in adverse conditions such as nutrient deprivation, DNA damage, or oxidative stress [37]. Resultant impaired rRNA synthesis and alterations in ribosome formation can accompany nucleolar stress, triggering significant downstream alterations in the nucleolus. Ultimately, overall cellular function is altered by regulating a wide spectrum of cellular processes via selected kinases or enzymes. Nucleolar stress is reported with a variety of signaling transductions, including Mdm2-p53, NF-κB, and HIF-1α pathways. For example, molecular changes from aberrant nucleolar function might activate p53, which further triggers cell cycle arrest or apoptosis. Aberrant nucleolar structure and/or function from internal or external cell stressors can divert downstream cell processes to a variety of cell fates, including apoptosis, senescence, autophagy, or differentiation [37,72,75,76]. These and additional transcriptional modifications in nociceptive pathways are important for the development and maintenance of pain associated with tissue damage [54,77,78,79,80]. We speculate from our work that the GluN1 subunit plays a role in these events based on the ultrastructural finding of GluN1 subunit staining along the nucleolar border (Figure 2). Nucleolar stress, such as nutrient deprivation, can manifest by increased nucleolar number and/or changes in nucleolar shape (Figure 3). Nucleolar stress may be precipitating the GluN1 subunit staining increases in nuclear regions observed in pain models and in vitro overactivation of cells with NMDA or NMDA plus ACPD. Multiple contributing stressors can impact nucleolar number, size, and functioning [81]. GluN1 subunit has been reported to interact with p53 [4], NFkB [82], and HIF-α [83] pathways, as shown in Table 2.

Table 2 is a summary of selected studies where the nuclear localization of the cells and reactions were reported. Columns are divided into Cellular Nuclear Regions, and Proteins reported to bind to the GluN1 subunit, Added reagents or stressors, Cellular or tissue responses due to the GluN1 subunit effect, and References. The nuclear regions mentioned are the Nuclear membrane, Nuclear pore complex (NPC), Chromatin, and Nucleolus. The first table column identifies the GluN1 subunit localization on the nuclear membrane or nuclear pore complex (NPC). Two major subnuclear organelles include the nucleolus and the chromatin, with loosely contained or densely contained DNA genetic material located liberally throughout the nucleoplasm. The nucleolus is an organelle lacking a lipid membrane, usually solitary, although >1 may be seen in an activated eukaryotic cell. The nucleolus is the site responsible for the biogenesis of ribosomes as part of DNA-to-RNA transcription. Additional models reflecting epigenetic long-term changes are reported [4,33,83]. 

## 6. Epigenetic Influence on GluN1 Subunit in Nuclear Organelles 

In the last 10 years, the concept of nucleolar stress has been described, defined, and thought responsible for directing cell machinery. Much has been reported on the involvement of GluN1 subunits in depression, seizures, psychosis, LTP, and memory. Depression and memory disruption are known to be a consequence of chronic pain. Epigenetic stress impacts nuclear chromatin and nucleolar operations of the cell. Pathways of relevance to epigenetic cell changes reported to impact GluN1 subunit expression and functions are histone deacetylase (HDAC) [86,87], NFkB/rel [82], and p53 [88]. DNA methylation, p53 activation, and post-transcriptional histone deacetylase are processes in the chromatin [86]. The GluN subunit has roles in cellular development and communication, reflecting dynamic cell retraction [34,67]. Early-life stress is a potent risk factor for the development of chronic pain, although the underlying mechanisms remain poorly understood. Studies of prenatal alcohol exposure in mice find the offspring have a greater risk for neuropathic pain, as one example [89,90]. Likewise, epigenetic changes and gene expression in the rat hippocampus are reportedly the response to novel environment exposure due to the interaction of D1/GluN subunits, phosphorylation of the NMDA and AMPA receptor subunits, and activation of ERK1/2 signaling [91]. 

## 7. Priming vs. Unprimed Target Activation: Or More of a Second Hit?

In addition to the impact of NMDA subunit heterogeneity, discrete cellular regions in oligodendritic cells in the brain have described glutaminergic differential physiologically active receptors, for example, AMPA receptors on the cell soma and GluN1 subunits on the oligodendrocyte processes. Pretreatment with MK801 prevented the excitotoxic effects on the processes normally seen in anoxic brain trauma [60,92]. 

The synopses below highlight the results from other studies and reinforce some observations: Priming vs. straight target activation. Many studies report the effects of long-term glutamate receptor activation. We also noted a much stronger, consistent downstream effect of GluN1 receptor activation in vitro when co-incubated with ACPD, or preincubated or co-incubated with PMA. Moreover, the timing or order of reagent exposure was important, in that PMA needed to be added first. In one set of studies, PMA was added in the first 6 h or last 6 h of the incubation. Much greater levels of TNF-α were detected in the cell culture supernatants of human clonal synoviocyte SW892 cell cultures if preincubated with nanomolar amounts of PMA [27]. Similarly, incubation with PMA is reported to prime the NFkB/Rel family in the nucleolus of non-neuronal HepG2 cells [65].

Priming exposure related to GluN1 subunit activation was reported in describing activation of BDNF as a contributor to colitis hypersensitivity, but low levels of BDNF provide protection from excitotoxicity [55,63,93,94]. Robust increases in TNF-α levels were evident when primary synoviocyte cultures derived from humans primed with active (inflammatory) synovitis were incubated with glutamate [95]. Co-incubation with phytohemagglutinin (PHA) and NMDA resulted in an increased global cell surface expression of GluN1 subunits and abrogation of interferon-gamma (IFN-γ) in the culture supernatant of lymphocytes [26]. The above suggests that at the level of the nucleolus, the ability of GluN1 subunit potential or membrane threshold changes that can direct changes to transcription and possibly cell fates is boosted or enhanced by priming or “first hit” to the nucleolus. Several reports describe cellular changes of weeks to months occurring from transcriptional diversions from the nucleolus [96,97].

## 8. Summary

This review was built on current hypotheses for some of these regulatory and pathologic functions, based on published subnuclear interactions and immunochemical imaging. Histological data demonstrating a shift in the subcellular location of the GluN1 subunit supports its role as a rapid intracellular mediator acting through direct communication with the nucleus and the nucleolus. Protein tyrosine kinase inhibitors can effectively reduce (i) pain-related behavior, (ii) GluN1 subunit expression increases in the spinal cord, and (iii) the shift of GluN1 subunit from a cell membrane to nuclear localization. Cycloheximide can block most of the glutamate-activated upregulation of GluN1 subunit content, confirming the synthesis of new protein in response to inflammatory insult contributions but not all content is necessarily de novo. 

Both the in vitro and the in vivo findings indicated that activated GluN1 subunits could produce rapid downstream events and result in increased GluN1 subunit protein potentially required to mobilize the shift in GluN2 subunit composition and receptor numbers that promote hypersensitivity in the acute K/C arthritis model. The modulation of trafficking to the cell nucleus promoting transcriptional changes would impact GluN1 subunit function and cellular response direction. Subsequently, the GluN1 subunit can stimulate, enhance, suppress, and de-repress via promoters, resulting in increased GluN1 subunit protein expression. Its transport onto chromatin in the nucleoplasm and the nucleoli allows the GluN1 subunit to respond quickly and consequently influence downstream nucleolar, nuclear, cellular, and functional processes. 

## Figures and Tables

**Figure 1 ijms-24-13196-f001:**
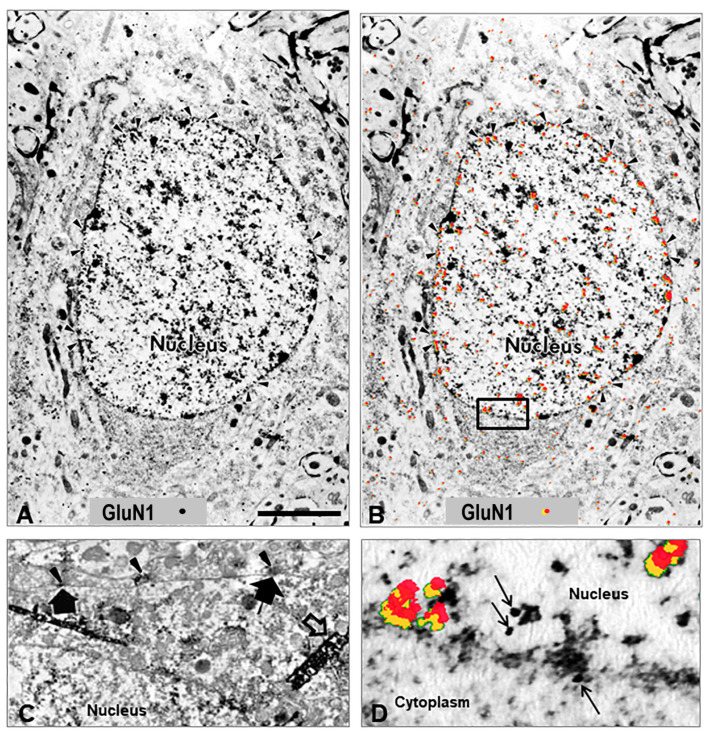
Nuclear localization of GluN1 subunit. (**A**) Electron micrograph illustrating the intracellular distribution of NMDA glutamate receptor GluN1 subunit in a spinal cord spinothalamic tract neuron 4 h after k/c induction of knee joint inflammation. The immunolocalization of the GluN1 subunit is evident in the nucleus at the rim of nuclear pores on the nuclear membrane rather than at the post-synaptic localization typical in spinal neurons of naive rats [45]. (**B**) The same EM image with red pseudocolor and yellow shadowing enhances the ability to observe the subcellular immunostaining of the colloidal gold particles found particularly in the nucleus, nuclear rim, and nuclear pores (arrowheads). (**C**) The GluN1 subunit in naïve animals is typically localized along the cell membrane. The arrowheads indicate the GluN1 subunit in the pre-synaptic region of terminals, and the large arrows indicate post-synaptic membrane immunoperoxidase labeling on a spinothalamic tract neuron identified by large dense crystals (open arrows) after WGA-HRP retrograde transport from the thalamus. (**D**) High power EM of the inset outlined in panel (**B**), with red pseudocolor (yellow shadowing) or arrows to indicate immunogold labeling of GluN1 subunit at the nuclear membrane and within the nucleus (n = 3). Scale Bar in (**A**,**B**) = 3 µm in panels; 1.33 µm in C; 0.3 µm in panel (**D**). Reprinted with copyright permission [40,45].

**Figure 2 ijms-24-13196-f002:**
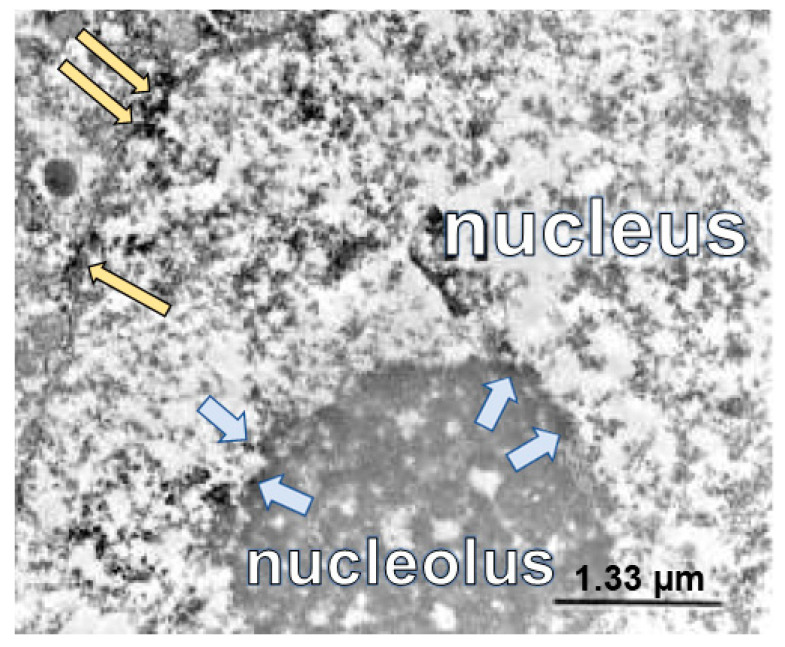
Nucleolar translocation of GluN1 subunit. Electron micrograph of GluN1 subunit immunolocalized with nickel intensified diaminobenzidine after K/C induction of knee joint inflammation in a rat. The immunoperoxidase labeling is localized on the nuclear pore edge and within the nucleus (long yellow arrows). The GluN1 subunit has migrated into the nucleus and is also localized at the edge of the nucleolus (shorter blue arrows). Labeled enlarged portion of an electron micrograph published previously, with permission [45].

**Figure 3 ijms-24-13196-f003:**
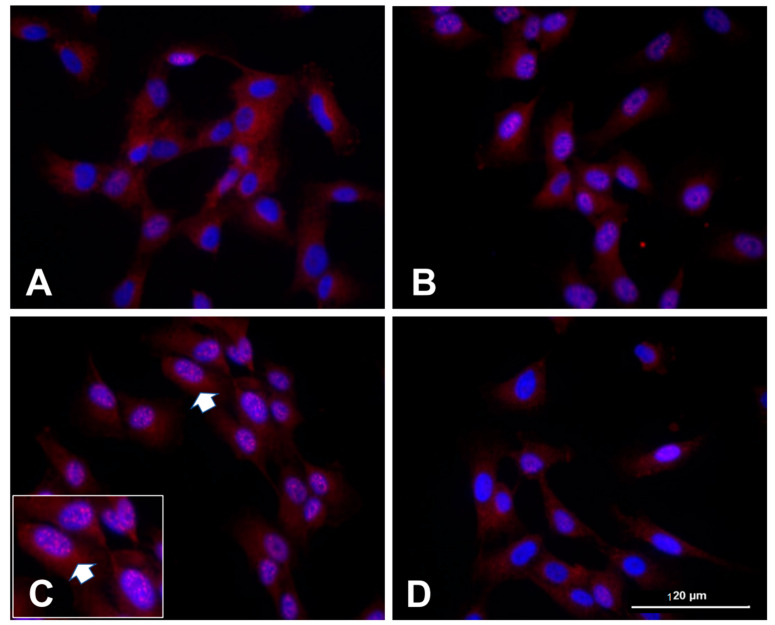
Human clonal synoviocytes with nuclear/nucleolar translocation of GluN1 subunit. Human synoviocytes (SW892), when untreated (**A**), displayed some cellular GluN1 subunit but no nuclear localization. Treatment of the synovial cultures with NMDA and ACPD for 1 h (**B**) and 3 h (**C**) increased nuclear GluN1 immunostaining over time. The nuclear immunostaining at 3 h is shown at higher power in the inset in (**C**) with an arrow indicating the same cluster of cells, each with multiple nucleoli surrounded by GluN1 subunit immunostaining. Multiple nucleoli in cells indicate high energy demands, ribosome biogenesis, and cellular stress. In panel (**D**), the synoviocytes were pretreated with MEK1/2 inhibitor U0126 (5 µM) prior to the treatment with NMDA and ACPD for 3 h, preventing the nuclear and nucleolar translocation. McNearney and Westlund, unpublished study, protocol per [27].

**Figure 4 ijms-24-13196-f004:**
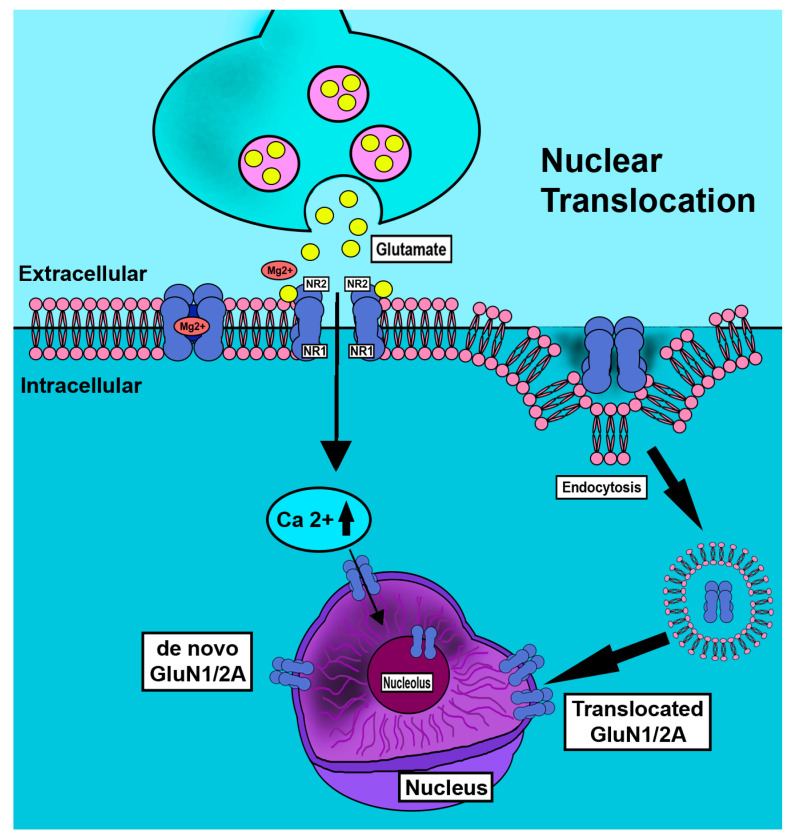
Schematic depiction of nuclear translocation of GluN1 subunit in neurons. Nociceptive neurons are overactivated in pain states. In conditions of abundance of increased GluN1 subunit is observed as increased cellular and nuclear ring immunostaining but is greatly reduced by tyrosine kinase inhibition [40]. Nuclear membrane translocation reportedly occurs by endocytotic [35] and de novo mechanisms.

**Table 1 ijms-24-13196-t001:** A brief synopsis of the studies highlighting results from various exposure protocols in experimental models of arthritis, formalin injection, colitis, and inflammatory bowel syndrome (IBS) that directly assess GluN1 subunit expression and function in pain states, cell cultures, tissue explants, animal models [9,19,23,24,27,31,40,41,49,55,56,57,58] and humans [27,51]. Studies demonstrating increased GluN agonist or inflammatory mediator levels, and preincubation with enhancing or conditioning agents are also included.

Model	Injury and/or Activation	Results	Pretreatment, Conditioning or Intervention	New Results after Pretreatment, Conditioning or Intervention	Reference
ARTHRITIS/ Inflammation					
Rat arthritis	Intraarticular (i.a.) kaolin/carrageenan (K/C) injection hind leg	Increased spinal dorsal horn (SCDH) neuron excitability	1. Ketamine 2. D-AP5 3. CNQX	Decreased spinal dorsal horn neuron hyperexcitability	Neugebauer et al., 1993 [49]
Rat arthritis	i.a., K/C injection hind leg	Increased 2–4-fold, i.a. Glu conc 10–120 min after K/C injection	Pretreat with i.a. CNQX, ketamine, lidocaine injection	Increased i.a. GLU conc. significantly blunted	Sluka and Westlund. 1993 [59] Lawand et al., 2000 [41]
Rat primary cerebellar granule neuron cultures	Glu incubation	Neurotoxicity	Pre-Rx with: 1. NMDA 2. MK-801 3. Transfected with DS oligonucleotide to BDNF NFkB binding site	Pretreatment and BDNF DNA binding activity resulted in neuroprotection	Lipsky et al., 2001 [55]
Mouse Lumbar SCDH (IPLformalin-induced nociception)	IPL micro-injection rAAV-GFP Lumbar SCDH IPL injection, formalin hind paw	Increased licking, biting pain behaviors Increased EPSC current and synaptic response in Lamina II neurons of SCDH	Conditional GluN1 deletion segments	Significant decrease in pain behaviors Significant decrease in EPSC’s by 60–80%	South et al., 2003 [9]
Rat (IPLformalin-induced nociception)	IPL injection, formalin hind paw	Increased number of hind paw flinches	GluN1 antisense mRNA	Significant blunting in the number of flinches	Lee et al., 2004 [56]
Rat Oligodendrocyte spines	Anoxic injury model in oligodendrocytes	GluN1 activation, Intracell Ca^2+^ activation; cell injury	Pretreat with CNQX or MK-801	GluR blockade blunted cell inactivation and injury—Specific GluN1 subcellular function by cell region	Salter and Fern, 2005 [60]
Rat arthritis	IPL injection, CFA hind paw	Nociceptive behavior, secondary mechanical allodynia	GluN1-coded siRNA-AAV vector injection into lumbar SCDH	Reduced pain behaviors by 60–75% for ≥6 mo	Garraway et al., 2007 [24]
Mouse spinal cord injury	Direct, blunt trauma to spinal cord motoneurons	Incr cell GluN1 perinuclear and nuclear staining	Add’n of PHA I at BsL, then NMDA	Increased GluN1 Staining	Mashkina et al., 2010 [26]
Rat arthritis	IPL injection CFA hind paw	Incr pGluN1 protein and spontaneous firing discharge rates in hypothalamic slices	Pre-injx MK801 or CNQX	Decreased pGluN1 levels and spontaneous firing	Peng et al., 2011 [61]
Rat Subcutaneous (s.c.) formalin-induced nociception	s.c. injection, formalin hind leg skin	Significant increase in number of paw flinches and in skin GluN1 staining	shRNA injection into the contralateral paw	Significant blunting of paw flinches and GluN1 staining	Tan et al., 2011 [58]
Mouse hippocampal neurons	NMDA added to hippocampal neurons and tissue slices		PreRx: neurons/slices transfect with GluN1 isoforms	Incr GluN1 C1 staining, most in neuronal cell nuclei -Incr EPSC activity	Zhou and Duan, 2018 [23]
Rat K/C arthritis	i.a. K/C injx hind leg	Secondary mechanical allodynia Incr SCDH GluN1, pGluN1, staining	Genistein Lavendustin A (PTK inhibitors)	Signif blunted pain behaviors, Signif blunted GluN1 staining	Westlund et al., 2020 [40]
COLITIS					
Rat Transient TNBS colitis	TNBS solution, intra-colonic	Increased GluN1 variants in spinal cord T10-L1 after colitis resolution			Zhou et al., 2009 [62]
Rat TNBS colitis	Day1 TNBS sol’n enema rectum to colon	D7: Incr pGluN1 in SCDH L1 and S1 D7: SCDH slices incubated w/BDNF-> increased pGluN1	D3: i.v. α-BDNF neutralizing Ab D7: slices pre-incubated P13/Akt, PLC-γ or PKC inhibition	D7: Colitis increased pGluN1 α-BDNF block decreased Western blot pGluN1 and pGluN1 immunostaining	Liu et al., 2015 [63]
Rat IBS model	1. Neonatal maternal separation, 2. Acetic acid enema 3. colorectal distension	Increased abdominal withdrawal reflex Decreased open-field activity test Incr GluN1 staining: RT-PCR in ACC; protein by IHC in the colon	Electroacupuncture on Neiguan points PC6 and ST36	-Decreased abdominal withdrawal reflex; Increased open field activity; Decreased GluN1 in ACC and colon	Tan et al., 2019 [64]
Non-neural cells, in vitro					
Rat, mouse osteoclasts	In vitro bone resorption model	GluN1 levels noted with bone resorption by osteoclasts	Pretreat with 1 D-AP5 2. MK-801 3. Anti GluN1 Ab	In vitro bone absorption was inhibited	Chenu et al., 1998 [31]
Rat arthritis, Human synoviocytes	NMDA, ACPD added to human clonal synoviocyte cultures, SW892	Increased GluN1 staining by ICC	Pre-incubated with microdose PMA	Significant increase in GluN1, TNF-α, and RANTES	McNearney et al., 2010 [27]
Other models—NFkB translocation					
Inflammation	HepG2 human liver cells Incubation with PMA	Increased levels of nuclear NFkB, Cytoplasmic COX-2,MMP-9	Pre-incubated with Protopine	Significant decreases in nuclear NFkB, cytoplasm COX-2, and MMP-9 levels	Kim et al., 2022 [65]

**Table 2 ijms-24-13196-t002:** Selected studies reporting GluN1 subunit subnuclear organelle locations.

Cellular Nuclear Regions	GluN1 Co-Expressed with	Expression System— Host Cells, Tissues	Added Reagents or Stressors	Cellular or Tissue Responses with NMDA NR1	References
Nuclear membrane	GluN2 GluN2 GluN3	Rat: primary neurons, dorsal and ventral horn tissue Human: clonal neuro-blastoma cells, SH5YSY Primary human synoviocyte cultures; Clonal human synoviocytes SW892 Clonal human synoviocytes, SW892 Human, mouse, rat brain tissue; Human clonal kidney cells, HEK-293	Kaolin/carrageenan (K/C) intra-articular (i.a.) rat arthritis model Glutamate Glutamate or NMDA PMA, NMDA, ACPD Elicited EPSC’s	EM: GluN1 increased at the nuclear membrane, secondary heat hyperalgesia Confocal: Increased nuclear and cytoplasmic GluN1 staining GluN1 staining: Increased at nuclear and cytoplasmic regions Whole-cell enlargement Increased supernatant levels: TNF-α, RANTES (Figure 3: Increased nucleoli number) The proposed neuroprotective modulator of GluN1 containing heterodimer	Westlund et al., 2020 [40] Westlund et al., 2020 [40] McNearney et al., 2010 [27] McNearney et al., 2010 [27] Figure 3 Tong et al., 2008 [84]
Nuclear Pore Complex	GluNA/B	Ipsilateral dorsal and ventral horns	Rat K/C arthritis model, K/C, i.a.	Nuclear access shuttle EM: GluN1 increases at nuclear membrane pores	Westlund et al., 2020 [40]
Chromatin	HIF-1	Rat brain PVN: (congestive heart failure model) Rat clonal neuronal cells NG108-15 * cells for hypoxia studies	Hypoxic stress, assess PVN for Glu neuro-excitation; NG108-15 cells for epigenetic changes	*HIF-1* binds to GluN1 promoter region; Increased HIF-1 and GluN1 levels, Increased cardiac excitation.	Sharma et al., 2016 [83]
Chromatin		Mouse–depression and anxiety models–study hippocampus	Acute restraint or forced swim stress on Chronic restraint- +/− increased P300 wave, BDNF levels	Increased GluN1 transcripts in the hippocampus after an acute forced swim on chronic restraint background. Changed behavior	Nasca et al., 2015 [33]
Chromatin	GluN2B	Rat, mouse, and human neuronal cells and brain slices	Incubated with added ethanol (ethanol)	Alcohol brain injury; decreased transcription; Increased histone modifications; DNA methylation; HDAC function, some long-term transcriptional changes	Reviewed in Chandrasekar, 2013 [4]
Nucleolus	GluN2A GluN3B	Human primary melanocytes Clonal melanoma cell lines	Plated and harvested at the confluence	N1/N2A subunits on primary cultures N1/N3B subunits on melanoma cells	Hajdu et al., 2018 [85]
Nucleolus	GluN3	Hydra. (*Hydra vulgaris*) nematoblasts, neuroblasts, and epitheliomuscular cells	Immunocytochemical staining with anti-fibrillarin ** and GluN1 Ab’s	Shuttle mechanism to nucleolus based on putative NoLS in 1 of 2 splice variants detected and GluN1 staining of nucleoli.	Kass-Simon et al., 2009 [67]
Nucleolus	Importin-α	Mouse primary hippocampal and cortical neuron cultures; Rat hippocampal slices	Cellular and tissue activation via electrical pulses	Importin-α binds to NLS of 3’ tail of cytoplasmic GluN1; GluN1 subunit activation releases importin-α for binding to proteins entering through the nuclear pore complex	Jeffrey et al., 2009 [21] ***

* NG108-15 is a hybrid cell line generated by fusing mouse N18TG2 neuroblastoma cells and rat C6-BU-1 glioma cells. ** Fibrillarin, is a component of a small nucleolar nuclear ribonucleoprotein particle localized to the nucleolus dense fibrillar component. *** Importin-α binds to cytoplasmic proteins and translocates to the cell’s nucleus [21].

## Data Availability

Not applicable.

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
