# Peer review of "Pluripotential GluN1 (NMDA NR1): Functional Significance in Cellular Nuclei in Pain/Nociception"

_ijms, 2023, doi:10.3390/ijms241713196_

Round 1

Reviewer 1 Report

The paper titled “Pluripotential GluN1 (NMDA NR1): Functional Significance in Cellular Nuclei in Pain/Nociception” is a very well-written review containing relevant information about the shift of the NR1of the plasma membrane to the nucleus.

The literature about the traffic of some membrane proteins to the nucleus is a field of research demonstrating the close relationship between the external inputs and the regulation of gene expression; furthermore, this paper highlights the diverse roles of the NMDA receptors: as an ion channel and as a direct putative regulator of the epigenetic.

I highly recommend this review for publication after a minor revision. 1.     In the following sentence the conjunction “and” seems use may be incorrect: “NMDA receptors and in low levels are essential for neuronal development” (lane 36, page 1. 2.     The following idea could be confusing to the reader. “NMDA receptors form as four subunit heterodimers with two obligate N1 subunits to anchor two N2, or NR3 subunits, comprising a functional ionic channel” (lane 51-52, page 2): I suggest the following change:  “Four subunits assemble the NMDA receptors, and each heterodimer strictly requires N1 subunit to anchor N2 or NR3 subunits…” In this paragraph, the authors could consider making a figure summarizing the overall structure of the NMAD receptors, which would help the reader to understand the structural arrangement of these receptors.
3.    
The following idea is also confusing:

The binding composition is noted as two obligatory GluN1 subunits of which there are now reported at least 8 distinct variant subunits and two variable subunits from the GluN2 (GluN2A-2D, four variable subunits) or GluN3 (GluN3A-3B, two variable subunits) subunit families (9x6=54 potential heterodimers). Page 2, lanes 57-60. The authors could consider simplifying the sentence, maybe as following: The binding composition is noted as two obligatory GluN1 subunits of which there are now reported at least eight distinct variant subunits and four or two variable subunits from the GluN2 (GluN2A-2D) and GluN3 (GluN3A-3B), respectively, producing 8x6=54 potential heterodimers. 4.     Please correct UG0126 (page 5, lane 180), it should read U0126. 5.     Please the authors could clarify what does mean: “GC rich regions cassette 1 of the GluN1 subunit”? page 9 lane 246? This region contains the bipartite NLS, which is rich in basic amino acids; please clarify. 6.     Please add the missing reference or correct: “p53 (XX)”, page 12 lane 339 7.     Please correct the symbol of TNF alpha and INF alpha in table 2 and page 13,lane 365, 371. 8.     Finally, the authors could be considered adding a figure illustrating the shift of GluN1 of the plasma membrane to the nucleus and the signals triggering it.

Author Response

Thank you for allowing us to improve our submission.

Reviewer 2 Report

The authors provided some evidence, mainly based on the histological data from authors' published papers, and made speculation that the nuclear NMDA receptor GluN1 subunit might play a role in nociception. In general, the manuscript provided a review in an organized matter concerning the possible function of nuclear GluN1, especially in pain. However, I have several concerns that should be addressed by the authors as follows.

1. The authors should mention the role of nuclear GluN1 in nociception in the Abstract.

2. Some content in the TEXT may need to be more relevant to the subchapter. For example, the content in the last two paragraphs (lines 114-132) in the subchapter "2.1" might not be related to "GluN1 signaling induces nuclear translocation". In this chapter, the authors should describe in more detail the regulatory mechanisms underlying the trafficking of GluN1.

3. The authors should check the correctness of Table 1. Table 1 lists the published papers examining GluN1 expression and function in pain states. However, there are some mistakes in Table 1, including (1) many references cited in Table 1 that did not examine the expression of GluN1. (2) several references cited in Table 1 did not add to the reference list.

4. The authors should re-check the reference citation in the TEXT and Reference. 

a.          Several references cited in the text are not shown in the reference list (for example, lines 301-302, 4 articles cited, but 3 are missing in the reference list, …)

b.          No citation indicated (line 339)

c.           Wrong citation (lines 482-484, I cannot find the reference in PubMed or Google ?) 

Other minor concerns:

  1. The term GluN1 or NR1 should be consistent in the TEXT. 
  2. "C,D" are not labeled in Figure 1. 
  3. Symbol characters need to be shown correctly in Table 2. 

Author Response

(The authors gave the same response as above.)

Reviewer 3 Report

The current review discussing the function and structure of GluN1 as the subunit of NMDA receptor and specifically that NR1 has major physiologic potential in tissue and/or subnuclear functioning assignments by focusing on pain/nociception. It is an interesting review on the role of NMDA receptor in neurological disorders, but there are some issues that are needed to be considered.

1. In the abstract there is no information about the content of the review, and what is the aim of the review. 

2. It is not clear why authors only focused on nociception?

3. In table 1, in the model column some animals were written rat and some Rat arthritis?

4. What is the difference between result and impact in table 1?

5. Some part of review authors mentioned GluN1 and in some parts NR1.

Author Response

(The authors gave the same response as above.)

Round 2

Reviewer 2 Report

No further comments.